# OpenReview forum: "Cross-View Lewis Weight Fusion Empowering Exemplar Replay for Federated Class-Incremental Learning"
_ICML.cc/2026/Conference — ICML 2026 regular_

### Official Review · Reviewer_oSR5 · 2026-03-05

**Soundness:** 3
**Presentation:** 4
**Significance:** 4
**Originality:** 4
**Overall Recommendation:** 5
**Confidence:** 5

**Summary:**

This paper introduces a Cross-view Lewis weight Fusion framework, termed CLIF, which is designed specifically for Federated Class-Incremental Learning. To strike a balance between model performance and privacy preservation during exemplar replay, CLIF leverages Lewis weights from multiple feature spaces to guide the selection of representative samples. The approach incorporates Cross-View Lewis Weight Fusion (CV-LWF) and Frequency-aware Weighted Training (FWT). Comprehensive experiments conducted on CIFAR-10/100 and Tiny-ImageNet, using various non-IID and client configurations, show that CLIF consistently outperforms competitive baseline methods by 1–6%.

**Compliance With Llm Reviewing Policy:**

Affirmed.

**Final Justification:**

The rebuttal has addressed all my concerns.

**Key Questions For Authors:**

It is observed in the experiments that the model achieves the optimal performance when the number of views is set to m=5, while the performance degrades when m=10. Could the authors explain with specific experimental results which part of the model is negatively affected by the redundant features introduced by an excessive number of views?

If the author can address my concerns, I will raise my score.

**Limitations:**

Yes

**Strengths And Weaknesses:**

Strengths:

1. The paper is well-organized and presents its ideas in a clear, logical sequence. The motivation is effectively communicated, making it easy for readers to understand and follow the content.

2. The cross-view Lewis weight fusion is original and innovative, contributing to the study on federated class-incremental learning.

3. Extensive experimental results are provided to thoroughly verify the effectiveness of the proposed method. Comprehensive ablation studies and analyses under various hyperparameter settings further validate its rationality and superiority.

Weaknesses:

1. In the experiments, CIFAR-10, CIFAR-100 and Tiny-ImageNet are split into different task sequences with varying numbers of phases, namely 3/5/5 and 5/10/10. Nevertheless, the motivation and rationale behind such experimental configurations are not fully elaborated.

2. In Section 3.3, the number of multi-views is defined as M, yet in subsequent experimental sections (e.g., Section 4.4 and Table 3), the symbol m is uniformly adopted to represent the number of views. This inconsistent use of symbols may easily cause confusion for readers when following the logical connection between the theoretical definition and experimental settings.

3. When the two variants of CLIF, namely CLIF_R and CLIF_F, are first introduced in Section 4.2, only their calculation and aggregation modes are briefly described without defining the full names corresponding to the suffixes (e.g., R for Local Replay and F for Global Fusion Replay). Readers have to infer the meanings by cross-referencing the context, which impairs the readability of the paper.

---

> ### Author Rebuttal · Authors · 2026-03-30
>
> **Response to W1**
>
> Thank you for this question. We use the task-splitting configurations on CIFAR-10, CIFAR-100, and Tiny-ImageNet for two main considerations.
>
> First, they allow us to evaluate the method under **different levels of incremental difficulty**. The shorter setting (3/5/5) corresponds to a relatively moderate continual-learning scenario, whereas the longer setting (5/10/10) creates a more demanding process with more phases and typically stronger forgetting. From the perspective of class allocation, CIFAR-100 and Tiny-ImageNet with 5/10 tasks correspond roughly to 20/10 (or 40/20) classes per phase, while CIFAR-10 with 3/5 tasks gives around 3–4 classes per phase. This keeps the per-phase difficulty in a comparable range while letting us examine robustness across different sequence lengths.
>
> Second, these choices are consistent with **common FCIL benchmarks in prior work**. For CIFAR-100 and Tiny-ImageNet, 5-task and 10-task settings are widely adopted in existing FCIL studies (FedCBDR, Re-Fed+, LANDER). For CIFAR-10, the 3-task and 5-task splits follow FedCBDR, which also makes our comparisons directly aligned with prior state-of-the-art results.
>
> **Response to W2**
>
> We thank the reviewer for pointing out this inconsistency. In the revised version, we will unify the notation for the number of views throughout the paper (we plan to consistently use **M**) and carefully check for similar notation inconsistencies.
>
>
> **Response to W3**
>
> We thank the reviewer for pointing out this readability issue. In the revised version, we will define their full names clearly at first mention and improve the description in Section 4.2. Specifically, **CLIF_R** denotes the variant built on the **local replay** pipeline, where cross-view Lewis weights are used for replay selection within each client’s local memory construction, while **CLIF_F** denotes the variant built on the **global replay** pipeline, where cross-view Lewis weights are used for global candidate selection on the server side. We will make this distinction explicit to improve readability.
>
> **Response to Q1**
>
> We thank the reviewer for this important question. The evidence we have mainly suggests that the impact occurs at the **replay-selection stage**: in **Table 3**, performance can slightly decline when the number of views increases from m=5 to m=10, indicating that additional views may introduce redundant/noisy features into the **cross-view Lewis-weight fusion**, leading to less informative exemplar ranking and selection. This is also consistent with **Appendix E.9**, where the benefit of CLIF is tied to improved sample diversity and representativeness.

---

> > ### Author Rebuttal · Reviewer_oSR5 · 2026-04-01
> >
> > Thanks for the authors' detailed responses. In general, the authors' rebuttal has satisfactorily addressed my concerns, specifically regarding the experimental setup and text descriptions. I have also carefully reviewed the other reviewers' comments and the authors' corresponding responses. I believe the authors have satisfactorily addressed the issues regarding the rationality of the key module design and privacy concerns. Overall, I am satisfied with the rebuttal and recommend acceptance with an increased score of 5.

---

### Official Review · Reviewer_itdm · 2026-03-06

**Soundness:** 4
**Presentation:** 3
**Significance:** 4
**Originality:** 4
**Overall Recommendation:** 5
**Confidence:** 2

**Summary:**

This work is on exemplar replay in Federated Class-Incremental Learning (FCIL), where clients learn new classes sequentially under non-IID data distributions while avoiding raw data sharing. Current exemplar-based methods use local-view selection,  lacking global awareness or use global-view selection, which incurs communication overhead and may leak feature statistics. To address this, the authors propose CLIF (Cross-view Lewis weight Fusion), a framework that selects replay exemplars via multi-view Lewis weight estimation. Each client constructs multiple feature “views” by using different client models (including other clients’ backbones) to extract representations. For each view, Lewis weights are computed, and the final sampling score is obtained via a max-fusion rule across views. This guarantees that the selected subset preserves the subspace embedding properties for all views simultaneously. Additionally, a frequency-aware weighted training module does exemplar re-weighting. They provide theoretical guarantees based on subspace embedding and operator norm preservation, arguing that the sampled exemplars preserve the Gram matrix and stabilize optimization. Experiments across multiple datasets, client counts, and heterogeneity settings show 1%–6% Top-1 accuracy gains over eight baselines. The paper also includes computational and memory overhead analysis.

**Compliance With Llm Reviewing Policy:**

Affirmed.

**Final Justification:**

The rebuttal has addressed most of the important concerns. I have upgraded the score for originality and retained my overall recommendation of Accept.

**Key Questions For Authors:**

1. Performance sensitivity to the number of views M: Is there a point where additional views introduce diminishing returns or instability?
2. Can you provide stronger empirical evidence that cross-view fusion meaningfully reduces privacy leakage compared to global-view feature sharing?
3. The claim of lower privacy risk than global-view feature sharing: Plausible but is it formally validated?

**Limitations:**

Yes

**Strengths And Weaknesses:**

Major Strengths:
-----------------------
1. Clear Problem Motivation: This work  addresses a well-defined gap in FCIL exemplar replay: balancing local representativeness, global awareness, privacy, and communication efficiency.
2. Principled Use of Lewis Weights: The adoption of Lewis weight sampling provides a strong theoretical backbone. The argument that overestimation-based sampling preserves subspace embeddings across views is technically sound and grounded in prior literature.

Other Strengths:
-----------------------
3. Cross-View Fusion Insight:  The max-based fusion of per-view Lewis weights is a simple and elegant design choice. It ensures simultaneous preservation across multiple representations and provides a clean theoretical justification.
4. Theoretical Analysis: The paper includes subspace embedding guarantees and bounds on prediction error under Lipschitz activation assumptions. The linkage between operator norm preservation and optimization stability is well motivated.
5. Empirical Coverage: Experiments cover multiple datasets (e.g., CIFAR, Tiny-ImageNet, DomainNet), different backbones (ResNet18, CLIP), multiple heterogeneity levels, and varying client numbers. They report 1%–6% improvement.
6. Overhead Analysis: Explicit runtime and GPU memory comparisons are included. Their analysis shows that increasing the number of views increases runtime but not peak GPU memory.

Weaknesses:
-------------------
1. Fusion Rule is not really a new Algorithmic Paradigm: While the application of Lewis weights is principled, the fusion rule (pointwise maximum) is relatively simple. The leap from single-view to cross-view looks more like an extension.
2. Strength of Theoretical Guarantees: The theoretical results establish subspace embedding and error bounds, based on assumptions like (full column rank, Lipschitz activation, sufficient sample complexity). How would they apply in deep nonlinear FCIL settings?
3. Privacy Claims Are Qualitative: The argument that cross-view replay avoids feature-statistics leakage inherent in global-view methods is primarily conceptual. There is no formal privacy analysis (e.g., differential privacy guarantees or attack simulations).
4. Communication Overhead Quantification:  Sending other-client backbones to construct views may incur non-trivial communication overhead. While mentioned, the trade-offs could be quantified more explicitly in the main paper.
5. Sensitivity to Undertrained Models:  The method requires that other clients’ models provide meaningful alternative views. If some client models are poorly trained or biased, cross-view fusion may amplify noise. This limitation is not deeply studied experimentally.
6. Limited Mechanistic Insight: Deeper analysis of decision boundary stability or class confusion patterns would help strengthen the empirical narrative.
--------
While my expertise in this area is low, overall it seems to me that the strengths outweigh weaknesses. The work is technically solid, theoretically motivated, and empirically validated. It makes a meaningful contribution to federated class-incremental learning.

---

> ### Author Rebuttal · Authors · 2026-03-30
>
> We appreciate the reviewer’s comments.
>
> **Response to W1**
>
> Our claim is not that this operator by itself forms a completely new paradigm, but that **it enables a nontrivial cross-view generalization of Lewis-weight replay in FCIL**. The key technical point is that **the fused score upper-bounds every view-specific Lewis weight, which makes the sampled subset simultaneously valid for all considered views and yields the guarantee in Theorem 1**. This is the main step beyond the single-view setting, where only one representation is protected.
>
> **Response to W2**
>
> We would like to clarify that **Theorem 1 provides a conditional representation-level guarantee for replay selection**. At any round in deep nonlinear FCIL, the client model induces feature representations through its nonlinear backbone. **Theorem 1 shows that CV-LWF selects exemplars that preserve all considered views simultaneously**, making replay on the selected subset a faithful surrogate for replay on the full old data in those feature spaces. Thus, the result applies task-wise to deep nonlinear FCIL by ensuring that the selected exemplars remain representative of the current deep representations used for downstream updates.
>
> **Response to W3/Q2/Q3**
>
> To provide stronger empirical evidence that cross-view fusion can reduce privacy leakage relative to global-view feature sharing, we further examine privacy risk from a reconstruction-attack perspective. In **Appendix E.8**, we propose **two alternative model-sharing strategies** as privacy-aware variants to mitigate leakage caused by cross-client model exchange. Moreover, in our response to **Reviewer iF7r (W1)**, we provide additional **DLG gradient inversion attack** experiments. The results show that these two alternatives substantially reduce the reconstruction-based privacy leakage risk, while causing only marginal changes in the final performance. We will revise the paper to make this empirical privacy comparison more explicit.
>
> **Response to W4**
>
> We agree that the communication trade-off should be stated more explicitly in the main paper. We already quantify this in the current submission: ** Table 4** reports runtime/memory overhead, and **Table 9** gives the communication formulas. We will move this analysis into the main paper to make the trade-off clearer.
>
> **Response to W5/Q1**
>
> In fact, the results already indicate that **increasing the number of views does not always improve performance**. In **Table 3**, using m=10 views can be slightly worse than using m=5 in some settings, indicating that additional client models may introduce redundant or noisy features. We therefore identify robustness to undertrained or biased client models as an important direction in the paper’s **Future Work**. Moreover, in **Appendix E.11**, we discuss possible **model-selection strategies**, e.g., selecting views based on local validation performance or feature similarity/diversity, to filter less reliable views before fusion.
>
>
> **Response to W6**
>
> We thank the reviewer for this valuable suggestion. We agree that deeper mechanistic analysis would further strengthen the empirical narrative. In fact, we have conducted **case-based error analysis**, including **decision boundary stability analysis**, to better understand how CLIF improves replay quality and mitigates forgetting. These results provide additional mechanistic support beyond the aggregate accuracy numbers. Since the rebuttal does not allow us to include additional figures, we are unable to present these visual analyses here. We will therefore add this analysis as a dedicated experimental subsection in the revised version, together with the corresponding figures and discussion.

---

> > ### Author Rebuttal · Reviewer_itdm · 2026-04-03
> >
> > The rebuttal effectively addresses several of my concerns:
> > •	The role of the max-fusion rule is clarified precisely, showing that it upper-bounds view-specific Lewis weights and enables simultaneous subspace preservation across views.
> > •	The theoretical guarantees are better contextualized for deep nonlinear FCIL, operating at the representation level induced by learned backbones.
> > •	The authors provide additional empirical evidence on privacy via gradient inversion attacks and propose mitigation strategies, strengthening the privacy-related claims.
> > •	Communication and computational trade-offs are clarified and to be made more explicit in the main paper.
> > Some limitations remain, not critical (e.g., incremental nature of the fusion rule, limited mechanistic analysis currently visible). Overall, the rebuttal strengthens this contribution and confirms my initial assessment.
> >
> > So, my overall recommendation is unchanged: Accept
> >
> > Additional Question:
> >
> > 1. Is the max-fusion operator essential for the theoretical guarantees, or would alternative fusion rules yield similar multi-view preservation properties?

---

> > > ### Author Response · Authors · 2026-04-03
> > >
> > > We thank the reviewer for this insightful follow-up question. We address it as follows.
> > >
> > > The key property exploited by Theorem 1 is that the fused score must **upper-bound** every view-specific Lewis weight, i.e.,
> > > $$
> > > \tilde{w}_{k,i} \geq w_i^{(p,m)}(A_k^{(m)}), \quad \forall\, m \in [M].
> > > $$
> > > The pointwise maximum is the **tightest** such upper bound that can be constructed from the per-view Lewis weights alone: it requires no assumptions about the relationships among the views (e.g., no shared spectral structure, no correlation between client models, no distributional assumptions on model parameters). This is particularly important in the FCIL setting, where client models are trained under heterogeneous, non-IID data and may exhibit highly diverse feature geometries. Any alternative that also upper-bounds all per-view weights (e.g., a weighted sum with sufficiently large coefficients, or the $\ell_q$-norm of the weight vector across views) would equally preserve the theoretical guarantee, but would necessarily be *looser* than the max, leading to a larger total weight
> > >
> > > $$
> > > T_k = \sum_i \tilde{w}_{k,i}
> > > $$
> > >
> > > and thus a larger sample complexity bound in Theorem 1.
> > >
> > > One could envision fusion rules that exploit structural relationships among views. For example, if the feature matrices $\{A_k^{(m)}\}_{m=1}^M$ share a common low-rank subspace or exhibit bounded pairwise spectral distances, it might be possible to derive a joint Lewis weight that is tighter than the pointwise max. However, establishing such bounds for a collection of deep neural network backbones, which are highly non-linear, trained under different data distributions, and may not share any convenient algebraic structure, remains an open theoretical challenge. To the best of our knowledge, no existing result provides such joint Lewis weight guarantees for heterogeneous nonlinear models.

---

### Official Review · Reviewer_rpaM · 2026-03-10

**Soundness:** 3
**Presentation:** 4
**Significance:** 4
**Originality:** 4
**Overall Recommendation:** 5
**Confidence:** 5

**Summary:**

This paper proposes a novel federated class-incremental learning framework named CLIF, which introduces multi-view representation and cross-view Lewis weight sampling to improve the quality of exemplar selection. By integrating local and global variants, the method effectively alleviates catastrophic forgetting and enhances model robustness under non-IID data distributions. Extensive experiments on multiple benchmark datasets demonstrate that CLIF achieves competitive performance compared with state-of-the-art methods. Meanwhile, lightweight privacy-preserving modules and reasonable computational overhead make the approach suitable for real-world edge federated learning scenarios.

**Compliance With Llm Reviewing Policy:**

Affirmed.

**Final Justification:**

The authors have adequately addressed the concerns raised by multiple reviewers regarding the experiments and algorithmic mechanisms. I maintain my recommendation of Accept.

**Key Questions For Authors:**

It is recommended that the author address the aforementioned weaknesses.

**Limitations:**

Yes, the limitations has been discussed in the future work.

**Strengths And Weaknesses:**

## Strengths:

- CLIF innovatively integrates cross-view Lewis weight fusion into exemplar replay, which effectively mitigates the single-view bias in local selection and the privacy risks in global feature sharing, achieving a well-balanced trade-off between exemplar representativeness and data privacy protection.

- The proposed Frequency-aware Weighted Training module optimizes the utilization of selected exemplars by modulating loss contributions based on cross-view selection frequency, which further strengthens the model's ability to alleviate catastrophic forgetting.

- The method is theoretically grounded with rigorous proof of subspace embedding guarantees for cross-view sampling, providing solid theoretical support for the rationality of the framework design rather than just empirical validation.

- Extensive experiments on multiple datasets and heterogeneous settings show the method outperforms SOTA baselines by a consistent 1%–6%, with its CLIF_R and CLIF_F variants adapting to local/global replay scenarios and demonstrating strong scalability and robustness. It also adopts lightweight privacy-preserving strategies with negligible performance loss, and its quantified modest computational overhead makes it well-suited for practical edge federated deployment.

## Weaknesses:

- The paper fails to specify the interaction mechanism between the local client model and other models during the calculation of cross-view Lewis weights, such as whether the complete model parameters or only the feature extraction layers are transmitted among clients.

- The experiments have demonstrated the performance differences between CLIF_R​ and CLIF_F under various experimental settings, yet a systematic generalization and clear definition of their applicable scenario boundaries have not been conducted.

- The experimental settings of hyperparameter ranges (e.g., local training epochs {1,2,4,8} and memory buffer sizes {20,40,80}) lack explicit illustrations of their selection basis, such as whether they follow mainstream configurations in the field or are verified by pre-experiments. This results in insufficient detailed support for the rationality of the experimental design of some hyperparameters.

---

> ### Author Rebuttal · Authors · 2026-03-30
>
> We appreciate the reviewer’s comments.
>
> **Response to W1**
>
> We thank the reviewer for pointing out this ambiguity. In our implementation, **only the feature encoder, not the complete model, is transmitted** for cross-view Lewis weight computation. Each client receives k additional encoders, applies them to its **own local data** to build multiple feature matrices, computes per-view Lewis weights, and then fuses them across views; the prediction heads are not needed in this step. This is also consistent with **Appendix E.7 / Table 9**, which quantifies the extra communication as k|θ_{enc}|. We will clarify this mechanism more explicitly in the revised paper.
>
> **Response to W2**
>
> We thank the reviewer for this helpful suggestion. In our framework, **CLIF_R** is better suited to settings where privacy and communication efficiency are more important, since it follows the **local replay** mechanism, whereas **CLIF_F** is more suitable when centralized replay is acceptable and slightly higher accuracy is desired, since it performs **global candidate ranking** on the server. Our experiments already reflect this trade-off, but we will make it more explicit in the revised paper.
>
> **Response to W3**
>
> We thank the reviewer for this helpful comment. Our intention was to use these settings as representative sensitivity ranges that cover **light to relatively strong local updates** {1,2,4,8} epochs) and **small to moderate replay budgets** {20,40,80} samples per task), so as to examine whether the relative behavior of the compared methods remains stable under different training and memory conditions. As shown in **Figure 4** and **Figure 5**, these ranges already reveal clear and consistent trends: performance generally improves with more local epochs or larger buffers, while gains become marginal or may slightly decline at larger local epochs, and the CLIF variants consistently outperform their corresponding baselines across these settings. We will revise the paper to clarify this selection rationale more explicitly in the experimental setup.

---

> > ### Author Rebuttal · Reviewer_rpaM · 2026-04-03
> >
> > After reviewing all the comments and the authors' rebuttal, I agree with reviewer itdm's conclusion that "It makes a meaningful contribution to federated class-incremental learning." I have no further questions. Given that this work is technically sound, theoretically motivated, and empirically verified, I am willing to maintain a positive score.

---

### Official Review · Reviewer_iF7r · 2026-03-13

**Soundness:** 3
**Presentation:** 3
**Significance:** 2
**Originality:** 2
**Overall Recommendation:** 3
**Confidence:** 3

**Summary:**

The paper studies federated class-incremental learning, where clients must learn new classes over time without forgetting old ones, while keeping data local. It proposes CLIF, a replay-based method that selects old samples using cross-view Lewis weight fusion, meaning each sample is scored from multiple client-model feature views so the replay buffer better preserves the geometry of the original feature space. It then uses frequency-based weighted training to give more importance to samples that are repeatedly identified as critical across views. Overall, the paper argues that this provides a better balance between replay quality, privacy, and communication than purely local or fully global replay strategies, and reports consistent gains over prior baselines on several benchmarks.

**Compliance With Llm Reviewing Policy:**

Affirmed.

**Final Justification:**

My main concern is somewhat deeper. In its current form, the paper does not adequately justify how cross-client model sharing fits with the privacy narrative it strongly relies on. The short experiment added in the rebuttal is helpful, but it is not enough to comprehensively support that claim. In particular, the added DLG-style attack result evaluates only one reconstruction-based threat, and therefore remains too limited to justify a broad privacy claim. While prior works can certainly help motivate this concern, I do not think citing them alone is sufficient here; since privacy is one of the paper’s central claims, this aspect needs to be supported more directly through the paper’s own experimental evidence. For this reason, I feel the paper may require further revision to more convincingly and comprehensively establish this part of its contribution. I am aware that other reviewers have given higher scores, and I recognize that some of their concerns may have been addressed in the rebuttal. However, my concern is different in nature and remains unresolved in my view, especially since privacy is one of the central motivations of the work. Therefore, I am keeping my original score.

**Key Questions For Authors:**

Please address the weaknesses section.

**Limitations:**

yes

**Strengths And Weaknesses:**

Strengths:
The paper has a clear and intuitive core idea: it aims to keep replay samples that best preserve the geometric structure of the original feature space, so replay training remains closer to training on the full old dataset

A key strength is that the method sits at a nice middle ground between purely local and fully global replay strategies. Instead of relying only on one local view, or requiring global feature sharing, it uses multiple client-model views to obtain a more robust estimate of sample importance.

The experimental section is extensive: the authors include multiple datasets, different heterogeneity settings, client counts, ablations, and case studies


Weaknesses:
Although privacy is the major motivation of the paper, the privacy story is still not fully convincing. The method avoids direct global feature sharing, but it still relies on transferring client models across participants, which can itself leak information about local data. The proposed mitigations are mostly heuristic, such as Gaussian perturbation, without any direct privacy evaluation against attacks. This is important because recent work such as [1], has shown that an active membership inference attack can still achieve high success even under rigorous local differential privacy protection, and it explicitly says that adding enough privacy-preserving noise to stop the attack would significantly hurt model utility. Because of this, the paper’s privacy advantage feels more intuitive than demonstrated: it reduces one obvious exposure route, but it does not quantify how much privacy risk remains under cross-client model exchange.

One of the major components of the work is FWT. Although FWT is empirically helpful, its methodological novelty appears limited, since the weighting rule and reweighted optimization are largely inherited from prior work [Huang et al. 2024]. This paper just renames it as a major module.

Their theorem is proved for an l_{p} -style objective, but the actual experiments use standard multi-class classification with cross-entropy. The authors do acknowledge this gap and say the same sampling rule is “naturally” applied to cross-entropy, based partly on prior empirical validation. Nevertheless, it makes the theoretical contribution slightly weaker. They also admit that the theoretical bound on replay size is much larger than the replay budgets used in practice.

Compared with simpler replay baselines, CLIF requires multiple feature extraction passes across different views as well as an additional weighting step, which naturally increases computational complexity. Although Table 4 suggests that the overhead is comparable to a few selected baselines, the comparison is still limited and does not cover a broader range of existing methods. Moreover, while CLIF generally improves performance, the gains are not consistent across all settings. The clearest improvements appear on CIFAR10, whereas on more challenging benchmarks such as Tiny-ImageNet, the margins are noticeably smaller.

The paper is motivated not only by accuracy, but also by privacy and communication efficiency. While Figure 1 claims that CLIF avoids the communication cost of global-view feature sharing, Table 4 does not report actual communication overhead, and Appendix E.7 provides only symbolic upload/download formulas rather than measured bandwidth or runtime. These formulas are useful for showing scaling trends, but they are not enough to fully support the communication-efficiency claim. Since this tradeoff is central to the paper, it would be more convincing to report concrete communication costs under the experimental setup, especially in comparison with the closest baselines such as Re-Fed+ and FedCBDR.

[1] Nguyen et al., “Active Membership Inference Attack under Local Differential Privacy in Federated Learning,” AISTATS 2023.

---

> ### Author Rebuttal · Authors · 2026-03-30
>
> We appreciate the reviewer’s comments.
>
> **Response to W1**
>
> We would like to clarify that this paper **does not overlook the privacy leakage risks arising from cross-client model sharing**. To further mitigate this issue, we propose two alternative approaches in Appendix E.8. These strategies are not merely heuristic conjectures and **they are practical mitigation mechanisms built upon common ideas of parameter perturbation and model combination from prior studies** [1,2]. More importantly, we further conducted a direct attack evaluation. Specifically, we performed the **DLG gradient inversion attack** [3] on the original shared parameters, parameters with Gaussian perturbation (+G), and parameters after Beta-weighted combination (+B), and measured the attack effectiveness using the PSNR between the reconstructed images and the original images. The results show that the PSNR under the original setting is 16.58, while it drops to 10.34 and 11.28 for the +G and +B schemes, respectively. These results demonstrate that **both alternative schemes make it substantially harder for attackers to recover the original images, thereby significantly reducing the reconstruction-based privacy leakage risk caused by cross-client model exchange**.
>
> | Metric| Ori| +G| +B |
> |-|-|-|-|
> | PSNR |16.58|10.34|11.28|
>
> [1] Fedexg: Federated learning with model exchange
>
> [2] Personalized federated learning with differential privacy and convergence guarantee
>
> [3] Clients collaborate: Flexible differentially private federated learning with guaranteed improvement of utility-privacy trade-off
>
> **Response to W2**
>
> We would like to clarify that **FWT is a training component tightly coupled with our cross-view fused sampling**. Specifically, we first construct fused Lewis scores from multiple client/model views to obtain a unified sampling distribution, and then perform weighted training on replay samples using the reweighted sampling matrix induced by this distribution. **This design allows the sampled exemplar subset to better preserve the subspace structure of the original feature space across different views**, and it naturally feeds into our theoretical analysis. Therefore, **the value of FWT does not lie in proposing a standalone new weighting formula, but in forming a unified framework together with cross-view fusion and Theorem 1**. The results in Appendix E.10 also indicate that FWT is not optional: compared with Focal Loss and class-balanced reweighting, **FWT performs better in most settings, suggesting that aligning training weights with sampling frequencies indeed helps mitigate catastrophic forgetting**.
>
> **Response to W3**
>
> We would like to clarify that the main role of our theory is **not to provide a one-to-one bound for the cross-entropy training used in experiments**. Rather, it is meant to show that **the fused Lewis-weight sampling rule preserves the multi-view subspace structure and provides the basis for reweighted replay**. As noted in the paper, **the guarantee is primarily qualitative**, and the replay size bound should be interpreted as a **sufficient theoretical condition**, rather than a prescription for the practical buffer budget.
>
> **Response to W4**
>
> As shown in the following table, **we have further expanded Table 4 to cover a broader range of existing data replay or generation based methods**. Moreover, existing methods (LANDER, FedCBDR) also tend to show more moderate improvements on Tiny-ImageNet. This is mainly because Tiny-ImageNet is inherently more challenging, with more classes, finer-grained visual differences, and stronger cross-task interference, leaving replay-based methods with more limited room for improvement. Even in this setting, CLIF still outperforms its corresponding baselines in most cases, showing stable benefits from the cross-view mechanism despite smaller gains.
>
> |Method|CIFAR10 Time (s)|CIFAR10 GPU Mem (MB)|Tiny-ImageNet Time (s)|Tiny-ImageNet GPU Mem (MB)|
> |-|-|-|-|-|
> |Target|49.360|3580|542.23|3892|
> |FCIDF|7.97|539|10.564|582|
>
> **Response to W5**
>
> We further provide the concrete communication cost statistics of all compared methods under the current experimental setup, as shown in the table below.
>
> |Method|Total Download|Total Upload|
> |-|-|-|
> |Re-Fed+ (Local replay)|44.59MB×n|44.59MB×n|
> |CLIF_R (Cross-view replay)|44.59MB×n+42.64MB×k|44.59MB×n|
> |FedCBDR(Global replay)|44.59MB×n+0.00049MB|44.59MB×n+3.91MB|
> |CLIF_F(Global cross-view replay)|44.59MB×n+42.64MB×k+0.00049MiB|44.59MB×n+3.91 MB×k|

---

> > ### Author Rebuttal · Reviewer_iF7r · 2026-04-03
> >
> > Thanks for your answers. However, I still find the privacy story somewhat insufficient for a paper that places such strong emphasis on privacy as a motivation. The added attack result is helpful, but it remains limited in scope. In particular, the rebuttal only evaluates a DLG-style gradient inversion attack, which provides evidence against one specific reconstruction-based threat, but is not enough to establish broad privacy protection under cross-client model exchange. As a result, the privacy claim still feels more suggestive than rigorously demonstrated.
> >
> > In addition, I am still not fully convinced about the paper’s methodological novelty. Both CV-LWF and FWT seem to be heavily built on prior Lewis-weight based machinery, and the rebuttal mainly emphasizes the value of combining these components within a unified framework rather than clarifying what is fundamentally new at the method level.

---

> > > ### Author Response · Authors · 2026-04-04
> > >
> > > We thank the reviewer for the follow-up. Our privacy motivation is **not based solely on intuition, but grounded in existing literature**. Prior work has shown that sharing intermediate / pseudo feature representations can itself leak substantial information about the original data [1,2]. For example, recent reconstruction attacks in split learning demonstrate that feature-level representations can serve as a direct recovery channel for private inputs. Therefore, our concern with global-view feature sharing is supported by prior evidence rather than intuition alone.
> > >
> > > To avoid explicit cross-client feature sharing, our method adopts the more common alternative of model-parameter exchange, which is widely used in federated learning [3,4,5]. At the same time, we fully recognize that model/gradient sharing may still leak private information, as classical gradient inversion work such as DLG has shown [6]. This is exactly why we further introduce alternative sharing strategies and empirically evaluate them against a DLG-style attack: the added results show that these alternatives substantially reduce reconstruction-based leakage risk while preserving similar final performance. More broadly, our privacy argument is also consistent with the standard view that **differential privacy provides stronger, formal protection against a broad class of privacy attacks**, whereas pseudo-feature sharing is at best a heuristic mitigation strategy. NIST’s 2025 guidelines explicitly describe differential privacy as a mathematical framework for quantifying privacy risk in machine learning and discuss it as a defense against privacy harms such as re-identification and extraction-related risks [7]. In this sense, our claim is not a speculative privacy story: it is based on **(i) prior evidence that pseudo/intermediate features could leak information, (ii) broad prior recognition that DP-style mechanisms provide stronger privacy protection, and (iii) our added attack experiments showing reduced leakage risk in practice**.
> > >
> > > [1] Efficient Heterogeneity-Aware Federated Active Data Selection
> > >
> > > [2] A Stealthy Wrongdoer: Feature-Oriented Reconstruction Attack against Split Learning
> > >
> > > [3] Personalized federated learning with first order model optimization
> > >
> > > [4] Adapt to adaptation: Learning personalization for cross-silo federated learning
> > >
> > > [5] Cross-training with prototypical distillation for improving the generalization of federated learning
> > >
> > > [6] Clients collaborate: Flexible differentially private federated learning with guaranteed improvement of utility-privacy trade-off
> > >
> > > [7] Guidelines for Evaluating Differential Privacy Guarantees
> > >
> > >
> > > Regarding novelty, our goal is not to claim a brand-new primitive in isolation, but to make a **meaningful contribution to the federated class-incremental learning (FCIL) community** by identifying how this line of machinery can be adapted to a setting where replay must operate under **cross-client heterogeneity, cross-view inconsistency, and continual forgetting**.
> > >
> > > More specifically, our methodological novelty lies in extending Lewis-weight-based replay from the **single-view** setting to the **cross-view FCIL** setting by constructing a fused sampling rule that yields a subset simultaneously valid for all views, which is exactly the role of CV-LWF in **Theorem 1**. In addition, **FWT** is not an isolated add-on, but the training counterpart of the reweighted sampling design, turning replay frequency into loss weighting so that the sampled subset better matches the intended surrogate objective. Empirically, Table 2 shows that **CV-LWF and FWT each contribute, and their combination yields the most consistent gains**, suggesting that the method is more than a simple reuse of prior components. We will revise the paper to clarify this novelty boundary more precisely and to better emphasize our contribution to the **FCIL problem setting**, rather than overstating the novelty of the underlying machinery itself.

---

### Decision · Program_Chairs · 2026-04-30

**Decision:**

Accept (regular)

**Comment:**

This paper received a mixed but overall favorable assessment. The reviewers agreed that the problem is important for FCIL, the core idea of cross-view Lewis-weight fusion is sensible, the theory is relevant though limited in scope, and the empirical study is broad, with generally consistent gains over replay-based baselines. The main negative arguments were that the privacy claims are not fully established, the methodological novelty is more incremental than foundational, and some efficiency claims were initially under-quantified. The rebuttal resolved most presentation, mechanism, and communication questions, but only partially addressed the central privacy concern raised by one reviewer.